# ON THE RELATIONSHIP BETWEEN SELF-ATTENTION AND CONVOLUTIONAL LAYERS

**Jean-Baptiste Cordonnier, Andreas Loukas & Martin Jaggi**
École Polytechnique Fédérale de Lausanne (EPFL)
`{first.last}@epfl.ch`

## ABSTRACT

Recent trends of incorporating attention mechanisms in vision have led researchers to reconsider the supremacy of convolutional layers as a primary building block. Beyond helping CNNs to handle long-range dependencies, Ramachandran et al. (2019) showed that attention can completely replace convolution and achieve state-of-the-art performance on vision tasks. This raises the question: do learned attention layers operate similarly to convolutional layers? This work provides evidence that attention layers can perform convolution and, indeed, they often learn to do so in practice. Specifically, we prove that a multi-head self-attention layer with sufficient number of heads is at least as expressive as any convolutional layer. Our numerical experiments then show that self-attention layers attend to pixel-grid patterns similarly to CNN layers, corroborating our analysis. Our code is publicly available[1].

## 1 INTRODUCTION

Recent advances in Natural Language Processing (NLP) are largely attributed to the rise of the *transformer* (Vaswani et al., 2017). Pre-trained to solve an unsupervised task on large corpora of text, transformer-based architectures, such as GPT-2 (Radford et al., 2018), BERT (Devlin et al., 2018) and Transformer-XL (Dai et al., 2019), seem to possess the capacity to learn the underlying structure of text and, as a consequence, to learn representations that generalize across tasks. The key difference between transformers and previous methods, such as recurrent neural networks (Hochreiter & Schmidhuber, 1997) and convolutional neural networks (CNN), is that the former can simultaneously attend to every word of their input sequence. This is made possible thanks to the *attention mechanism*—originally introduced in Neural Machine Translation to better handle long-range dependencies (Bahdanau et al., 2015). With self-attention in particular, the similarity of two words in a sequence is captured by an attention score measuring the distance of their representations. The representation of each word is then updated based on those words whose attention score is highest.

Inspired by its capacity to learn meaningful inter-dependencies between words, researchers have recently considered utilizing self-attention in vision tasks. Self-attention was first added to CNN by either using channel-based attention (Hu et al., 2018) or non-local relationships across the image (Wang et al., 2018). More recently, Bello et al. (2019) augmented CNNs by replacing some convolutional layers with self-attention layers, leading to improvements on image classification and object detection tasks. Interestingly, Ramachandran et al. (2019) noticed that, even though state-of-the art results are reached when attention and convolutional features are combined, under same computation and model size constraints, self-attention-*only* architectures also reach competitive image classification accuracy.

*These findings raise the question, do self-attention layers process images in a similar manner to convolutional layers?* From a theoretical perspective, one could argue that transfomers have the capacity to simulate any function—including a CNN. Indeed, Pérez et al. (2019) showed that a multi-layer attention-based architecture with additive positional encodings is Turing complete under some strong theoretical assumptions, such as unbounded precision arithmetic. Unfortunately, universality results do not reveal how a machine solves a task, only that it has the capacity to do so. Thus, the question of how self-attention layers actually process images remains open.

---

[1]Code: `github.com/epfml/attention-cnn`. Website: `epfml.github.io/attention-cnn`.

**Contributions.** In this work, we put forth theoretical and empirical evidence that self-attention layers can (and do) learn to behave similar to convolutional layers:

I. From a theoretical perspective, we provide a constructive proof showing that self-attention layers can express any convolutional layers.

Specifically, we show that a single multi-head self-attention layer using relative positional encoding can be re-parametrized to express any convolutional layer.

II. Our experiments show that the first few layers of attention-only architectures (Ramachandran et al., 2019) do learn to attend on grid-like pattern around each query pixel, similar to our theoretical construction.

Strikingly, this behavior is confirmed both for our quadratic encoding, but also for relative encoding that is learned. Our results seem to suggest that localized convolution is the right inductive bias for the first few layers of an image classifying network. We provide an interactive website[2] to explore how self-attention exploits localized position-based attention in lower layers and content-based attention in deeper layers. For reproducibility purposes, our code is publicly available.

## 2 BACKGROUND ON ATTENTION MECHANISMS FOR VISION

We here recall the mathematical formulation of self-attention layers and emphasize the role of positional encodings.

### 2.1 THE MULTI-HEAD SELF-ATTENTION LAYER

Let $\boldsymbol{X} \in \mathbb{R}^{T \times D_{in}}$ be an input matrix consisting of $T$ tokens in of $D_{in}$ dimensions each. While in NLP each token corresponds to a word in a sentence, the same formalism can be applied to any sequence of $T$ discrete objects, e.g. pixels. A self-attention layer maps any query token $t \in [T]$ from $D_{in}$ to $D_{out}$ dimensions as follows:

$$\text{Self-Attention}(\boldsymbol{X})_{t,:} := \text{softmax}\left(\boldsymbol{A}_{t,:}\right) \boldsymbol{X} \boldsymbol{W}_{val}, \tag{1}$$

where we refer to the elements of the $T \times T$ matrix

$$\boldsymbol{A} := \boldsymbol{X} \boldsymbol{W}_{qry} \boldsymbol{W}_{key}^{\top} \boldsymbol{X}^{\top} \tag{2}$$

as *attention scores* and the softmax output[3] as *attention probabilities*. The layer is parametrized by a query matrix $\boldsymbol{W}_{qry} \in \mathbb{R}^{D_{in} \times D_k}$, a key matrix $\boldsymbol{W}_{key} \in \mathbb{R}^{D_{in} \times D_k}$ and a value matrix $\boldsymbol{W}_{val} \in \mathbb{R}^{D_{in} \times D_{out}}$. For simplicity, we exclude any residual connections, batch normalization and constant factors.

A key property of the self-attention model described above is that it is equivariant to reordering, that is, it gives the same output independently of how the $T$ input tokens are shuffled. This is problematic for cases we expect the order of things to matter. To alleviate the limitation, a *positional encoding* is learned for each token in the sequence (or pixel in an image), and added to the representation of the token itself before applying self-attention

$$\boldsymbol{A} := (\boldsymbol{X} + \boldsymbol{P}) \boldsymbol{W}_{qry} \boldsymbol{W}_{key}^{\top} (\boldsymbol{X} + \boldsymbol{P})^{\top}, \tag{3}$$

where $\boldsymbol{P} \in \mathbb{R}^{T \times D_{in}}$ contains the embedding vectors for each position. More generally, $\boldsymbol{P}$ may be substituted by any function that returns a vector representation of the position.

It has been found beneficial in practice to replicate this self-attention mechanism into *multiple heads*, each being able to focus on different parts of the input by using different query, key and value matrices. In multi-head self-attention, the output of the $N_h$ heads of output dimension $D_h$ are concatenated and projected to dimension $D_{out}$ as follows:

$$\text{MHSA}(\boldsymbol{X}) := \underset{h \in [N_h]}{\text{concat}} \left[ \text{Self-Attention}_h(\boldsymbol{X}) \right] \boldsymbol{W}_{out} + \boldsymbol{b}_{out} \tag{4}$$

and two new parameters are introduced: the projection matrix $\boldsymbol{W}_{out} \in \mathbb{R}^{N_h D_h \times D_{out}}$ and a bias term $\boldsymbol{b}_{out} \in \mathbb{R}^{D_{out}}$.

---

[2]`epfml.github.io/attention-cnn`
[3]$\text{softmax}\left(\boldsymbol{A}_{t,:}\right)_k = \exp(\boldsymbol{A}_{t,k})/\sum_p \exp(\boldsymbol{A}_{t,p})$

## 2.2 ATTENTION FOR IMAGES

Convolutional layers are the *de facto* choice for building neural networks that operate on images. We recall that, given an image tensor $\mathbf{X} \in \mathbb{R}^{W \times H \times D_{in}}$ of width $W$, height $H$ and $D_{in}$ channels, the output of a convolutional layer for pixel $(i, j)$ is given by

$$\text{Conv}(\boldsymbol{X})_{i,j,:} := \sum_{(\delta_1, \delta_2) \in \boldsymbol{\Delta}_K} \mathbf{X}_{i+\delta_1, j+\delta_2, :} \mathbf{W}_{\delta_1, \delta_2, :, :} + \boldsymbol{b}, \tag{5}$$

where $\mathbf{W}$ is the $K \times K \times D_{in} \times D_{out}$ weight tensor [4], $\boldsymbol{b} \in \mathbb{R}^{D_{out}}$ is the bias vector and the set

$$\boldsymbol{\Delta}_K := \left[ -\left\lfloor \frac{K}{2} \right\rfloor, \cdots, \left\lfloor \frac{K}{2} \right\rfloor \right] \times \left[ -\left\lfloor \frac{K}{2} \right\rfloor, \cdots, \left\lfloor \frac{K}{2} \right\rfloor \right]$$

contains all possible shifts appearing when convolving the image with a $K \times K$ kernel.

In the following, we review how self-attention can be adapted from 1D sequences to images.

With images, rather than tokens, we have query and key pixels $\boldsymbol{q}, \boldsymbol{k} \in [W] \times [H]$. Accordingly, the input is a tensor $\mathbf{X}$ of dimension $W \times H \times D_{in}$ and each attention score associates a query and a key pixel.

To keep the formulas consistent with the 1D case, we abuse notation and slice tensors by using a 2D index vector: if $\boldsymbol{p} = (i, j)$, we write $\mathbf{X}_{\boldsymbol{p},:}$ and $\mathbf{A}_{\boldsymbol{p},:}$ to mean $\mathbf{X}_{i,j,:}$ and $\mathbf{A}_{i,j,:,:}$, respectively. With this notation in place, the multi-head self attention layer output at pixel $\boldsymbol{q}$ can be expressed as follows:

$$\text{Self-Attention}(\boldsymbol{X})_{\boldsymbol{q},:} = \sum_{\boldsymbol{k}} \text{softmax} \left( \mathbf{A}_{\boldsymbol{q},:} \right)_{\boldsymbol{k}} \mathbf{X}_{\boldsymbol{k},:} \, \boldsymbol{W}_{val} \tag{6}$$

and accordingly for the multi-head case.

## 2.3 POSITIONAL ENCODING FOR IMAGES

There are two types of positional encoding that has been used in transformer-based architectures: the *absolute* and *relative* encoding (see also Table 3 in the Appendix).

With absolute encodings, a (fixed or learned) vector $\mathbf{P}_{\boldsymbol{p},:}$ is assigned to each pixel $\boldsymbol{p}$. The computation of the attention scores we saw in eq. (2) can then be decomposed as follows:

$$\begin{aligned}
\mathbf{A}_{\boldsymbol{q}, \boldsymbol{k}}^{\text{abs}} &= (\mathbf{X}_{\boldsymbol{q},:} + \mathbf{P}_{\boldsymbol{q},:}) \boldsymbol{W}_{qry} \boldsymbol{W}_{key}^\top (\mathbf{X}_{\boldsymbol{k},:} + \mathbf{P}_{\boldsymbol{k},:})^\top \\
&= \mathbf{X}_{\boldsymbol{q},:} \boldsymbol{W}_{qry} \boldsymbol{W}_{key}^\top \mathbf{X}_{\boldsymbol{k},:}^\top + \mathbf{X}_{\boldsymbol{q},:} \boldsymbol{W}_{qry} \boldsymbol{W}_{key}^\top \mathbf{P}_{\boldsymbol{k},:}^\top + \mathbf{P}_{\boldsymbol{q},:} \boldsymbol{W}_{qry} \boldsymbol{W}_{key}^\top \mathbf{X}_{\boldsymbol{k},:}^\top + \mathbf{P}_{\boldsymbol{q},:} \boldsymbol{W}_{qry} \boldsymbol{W}_{key}^\top \mathbf{P}_{\boldsymbol{k},:}
\end{aligned} \tag{7}$$

where $\boldsymbol{q}$ and $\boldsymbol{k}$ correspond to the query and key pixels, respectively.

The relative positional encoding was introduced by Dai et al. (2019). The main idea is to only consider the position difference between the query pixel (pixel we compute the representation of) and the key pixel (pixel we attend) instead of the absolute position of the key pixel:

$$\mathbf{A}_{\boldsymbol{q}, \boldsymbol{k}}^{\text{rel}} := \mathbf{X}_{\boldsymbol{q},:}^\top \boldsymbol{W}_{qry}^\top \boldsymbol{W}_{key} \mathbf{X}_{\boldsymbol{k},:} + \mathbf{X}_{\boldsymbol{q},:}^\top \boldsymbol{W}_{qry}^\top \widehat{\boldsymbol{W}}_{key} \, \boldsymbol{r}_{\boldsymbol{\delta}} + \boldsymbol{u}^\top \boldsymbol{W}_{key} \mathbf{X}_{\boldsymbol{k},:} + \boldsymbol{v}^\top \widehat{\boldsymbol{W}}_{key} \, \boldsymbol{r}_{\boldsymbol{\delta}} \tag{8}$$

In this manner, the attention scores only depend on the shift $\boldsymbol{\delta} := \boldsymbol{k} - \boldsymbol{q}$. Above, the learnable vectors $\boldsymbol{u}$ and $\boldsymbol{v}$ are unique for each head, whereas for every shift $\boldsymbol{\delta}$ the relative positional encoding $\boldsymbol{r}_{\boldsymbol{\delta}} \in \mathbb{R}^{D_p}$ is shared by all layers and heads. Moreover, now the key weights are split into two types: $\boldsymbol{W}_{key}$ pertain to the input and $\widehat{\boldsymbol{W}}_{key}$ to the relative position of pixels.

## 3 SELF-ATTENTION AS A CONVOLUTIONAL LAYER

This section derives sufficient conditions such that a multi-head self-attention layer can simulate a convolutional layer. Our main result is the following:

**Theorem 1.** *A multi-head self-attention layer with $N_h$ heads of dimension $D_h$, output dimension $D_{out}$ and a relative positional encoding of dimension $D_p \geq 3$ can express any convolutional layer of kernel size $\sqrt{N_h} \times \sqrt{N_h}$ and $\min(D_h, D_{out})$ output channels.*

---

[4]To simplify notation, we index the first two dimensions of the tensor from $-\lfloor K/2 \rfloor$ to $\lfloor K/2 \rfloor$.

The theorem is proven constructively by selecting the parameters of the multi-head self-attention layer so that the latter acts like a convolutional layer. In the proposed construction, the attention scores of each self-attention head should attend to a different relative shift within the set $\mathbb{\Delta}_K = \{-\lfloor K/2 \rfloor, \ldots, \lfloor K/2 \rfloor\}^2$ of all pixel shifts in a $K \times K$ kernel. The exact condition can be found in the statement of Lemma 1.

Then, Lemma 2 shows that the aforementioned condition is satisfied for the relative positional encoding that we refer to as the *quadratic encoding*:

$$\boldsymbol{v}^{(h)} := -\alpha^{(h)}\left(1, -2\boldsymbol{\Delta}_1^{(h)}, -2\boldsymbol{\Delta}_2^{(h)}\right) \quad \boldsymbol{r}_{\boldsymbol{\delta}} := (\|\boldsymbol{\delta}\|^2, \delta_1, \delta_2) \quad \boldsymbol{W}_{qry} = \boldsymbol{W}_{key} := \boldsymbol{0} \quad \widehat{\boldsymbol{W}_{key}} := \boldsymbol{I} \quad (9)$$

The learned parameters $\boldsymbol{\Delta}^{(h)} = (\boldsymbol{\Delta}_1^{(h)}, \boldsymbol{\Delta}_2^{(h)})$ and $\alpha^{(h)}$ determine the center and width of attention of each head, respectively. On the other hand, $\boldsymbol{\delta} = (\delta_1, \delta_2)$ is fixed and expresses the relative shift between query and key pixels.

It is important to stress that the above encoding is not the only one for which the conditions of Lemma 1 are satisfied. In fact, in our experiments, the relative encoding learned by the neural network also matched the conditions of the lemma (despite being different from the quadratic encoding). Nevertheless, the encoding defined above is very efficient in terms of size, as only $D_p = 3$ dimensions suffice to encode the relative position of pixels, while also reaching similar or better empirical performance (than the learned one).

The theorem covers the general convolution operator as defined in eq. (17). However, machine learning practitioners using differential programming frameworks (Paszke et al., 2017; Abadi et al., 2015) might question if the theorem holds for all hyper-parameters of 2D convolutional layers:

- *Padding*: a multi-head self-attention layer uses by default the `"SAME"` padding while a convolutional layer would decrease the image size by $K - 1$ pixels. The correct way to alleviate these boundary effects is to pad the input image with $\lfloor K/2 \rfloor$ zeros on each side. In this case, the cropped output of a MHSA and a convolutional layer are the same.

- *Stride*: a strided convolution can be seen as a convolution followed by a fixed pooling operation—with computational optimizations. Theorem 1 is defined for stride 1, but a fixed pooling layer could be appended to the Self-Attention layer to simulate any stride.

- *Dilation*: a multi-head self-attention layer can express any dilated convolution as each head can attend a value at any pixel shift and form a (dilated) grid pattern.

**Remark for the 1D case.** Convolutional layers acting on sequences are commonly used in the literature for text (Kim, 2014), as well as audio (van den Oord et al., 2016) and time series (Franceschi et al., 2019). Theorem 1 can be straightforwardly extended to show that multi-head self-attention with $N_h$ heads can also simulate a 1D convolutional layer with a kernel of size $K = N_h$ with $\min(D_h, D_{out})$ output channels using a positional encoding of dimension $D_p \geq 2$. Since we have not tested empirically if the preceding construction matches the behavior of 1D self-attention in practice, we cannot claim that it actually learns to convolve an input sequence—only that it has the capacity to do so.

PROOF OF MAIN THEOREM

The proof follows directly from Lemmas 1 and 2 stated below:

**Lemma 1.** *Consider a multi-head self-attention layer consisting of $N_h = K^2$ heads, $D_h \geq D_{out}$ and let $\boldsymbol{f} : [N_h] \to \mathbb{\Delta}_K$ be a bijective mapping of heads onto shifts. Further, suppose that for every head the following holds:*

$$\text{softmax}(\boldsymbol{A}_{\boldsymbol{q},:}^{(h)})_{\boldsymbol{k}} = \begin{cases} 1 & \text{if } \boldsymbol{f}(h) = \boldsymbol{q} - \boldsymbol{k} \\ 0 & \text{otherwise.} \end{cases} \quad (10)$$

*Then, for any convolutional layer with a $K \times K$ kernel and $D_{out}$ output channels, there exists $\{\boldsymbol{W}_{val}^{(h)}\}_{h \in [N_h]}$ such that $\text{MHSA}(\boldsymbol{X}) = \text{Conv}(\boldsymbol{X})$ for every $\boldsymbol{X} \in \mathbb{R}^{W \times H \times D_{in}}$.*

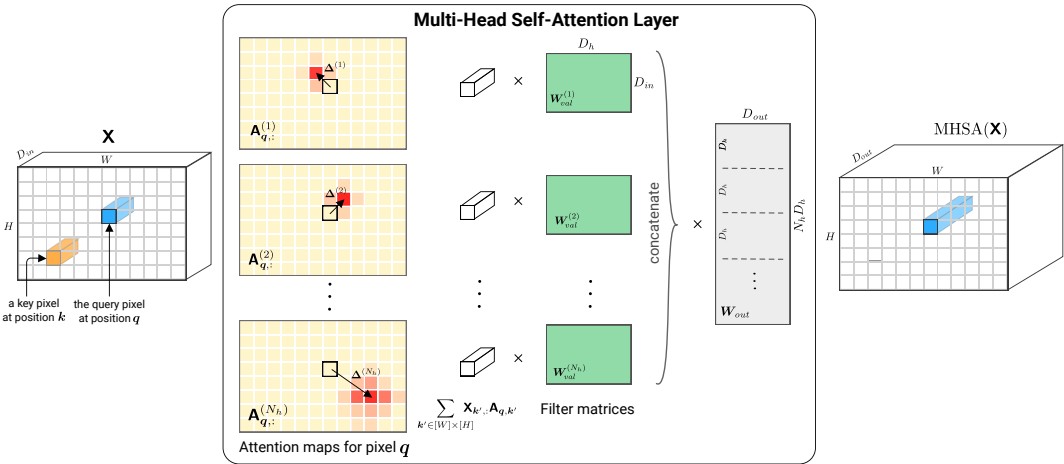

Figure 1: Illustration of a Multi-Head Self-Attention layer applied to a tensor image $\mathbf{X}$. Each head $h$ attends pixel values around shift $\mathbf{\Delta}^{(h)}$ and learn a filter matrix $\boldsymbol{W}_{val}^{(h)}$. We show attention maps computed for a query pixel at position $\boldsymbol{q}$.

*Proof.* Our first step will be to rework the expression of the Multi-Head Self-Attention operator from equation (1) and equation (4) such that the effect of the multiple heads becomes more transparent:

$$\text{MHSA}(\boldsymbol{X}) = \boldsymbol{b}_{out} + \sum_{h \in [N_h]} \text{softmax}(\boldsymbol{A}^{(h)}) \boldsymbol{X} \underbrace{\boldsymbol{W}_{val}^{(h)} \boldsymbol{W}_{out}[(h-1)D_h + 1 : hD_h + 1]}_{\boldsymbol{W}^{(h)}} \quad (11)$$

Note that each head's value matrix $\boldsymbol{W}_{val}^{(h)} \in \mathbb{R}^{D_{in} \times D_h}$ and each block of the projection matrix $\boldsymbol{W}_{out}$ of dimension $D_h \times D_{out}$ are learned. Assuming that $D_h \geq D_{out}$, we can replace each pair of matrices by a learned matrix $\boldsymbol{W}^{(h)}$ for each head. We consider one output pixel of the multi-head self-attention:

$$\text{MHSA}(\boldsymbol{X})_{\boldsymbol{q},:} = \sum_{h \in [N_h]} \left( \sum_{\boldsymbol{k}} \text{softmax}(\boldsymbol{A}_{\boldsymbol{q},:}^{(h)})_{\boldsymbol{k}} \boldsymbol{X}_{\boldsymbol{k},:} \right) \boldsymbol{W}^{(h)} + \boldsymbol{b}_{out} \quad (12)$$

Due to the conditions of the Lemma, for the $h$-th attention head the attention probability is one when $\boldsymbol{k} = \boldsymbol{q} - \boldsymbol{f}(h)$ and zero otherwise. The layer's output at pixel $\boldsymbol{q}$ is thus equal to

$$\text{MHSA}(\mathbf{X})_{\boldsymbol{q}} = \sum_{h \in [N_h]} \mathbf{X}_{\boldsymbol{q}-\boldsymbol{f}(h),:} \boldsymbol{W}^{(h)} + \boldsymbol{b}_{out} \quad (13)$$

For $K = \sqrt{N_h}$, the above can be seen to be equivalent to a convolutional layer expressed in eq. 17: there is a one to one mapping (implied by map $\boldsymbol{f}$) between the matrices $\boldsymbol{W}^{(h)}$ for $h = [N_h]$ and the matrices $\mathbf{W}_{k_1,k_2,:,:}$ for all $(k_1, k_2) \in [K]^2$. □

**Remark about $D_h$ and $D_{out}$.** It is frequent in transformer-based architectures to set $D_h = D_{out}/N_h$, hence $D_h < D_{out}$. In that case, $\boldsymbol{W}^{(h)}$ can be seen to be of rank $D_{out} - D_h$, which does not suffice to express every convolutional layer with $D_{out}$ channels. Nevertheless, it can be seen that any $D_h$ out of $D_{out}$ outputs of $\text{MHSA}(\boldsymbol{X})$ can express the output of any convolutional layer with $D_h$ output channels. To cover both cases, in the statement of the main theorem we assert that the output channels of the convolutional layer should be $\min(D_h, D_{out})$. In practice, we advise to concatenate heads of dimension $D_h = D_{out}$ instead of splitting the $D_{out}$ dimensions among heads to have exact re-parametrization and no "unused" channels.

**Lemma 2.** *There exists a relative encoding scheme $\{\boldsymbol{r}_{\boldsymbol{\delta}} \in \mathbb{R}^{D_p}\}_{\boldsymbol{\delta} \in \mathbb{Z}^2}$ with $D_p \geq 3$ and parameters $\boldsymbol{W}_{qry}, \boldsymbol{W}_{key}, \widehat{\boldsymbol{W}}_{key}, \boldsymbol{u}$ with $D_p \leq D_k$ such that, for every $\boldsymbol{\Delta} \in \mathbb{\Delta}_K$ there exists some vector $\boldsymbol{v}$ (conditioned on $\boldsymbol{\Delta}$) yielding $\text{softmax}(\boldsymbol{A}_{\boldsymbol{q},:})_{\boldsymbol{k}} = 1$ if $\boldsymbol{k} - \boldsymbol{q} = \boldsymbol{\Delta}$ and zero, otherwise.*

*Proof.* We show by construction the existence of a $D_p = 3$ dimensional relative encoding scheme yielding the required attention probabilities.

As the attention probabilities are independent of the input tensor $\mathbf{X}$, we set $\boldsymbol{W}_{key} = \boldsymbol{W}_{qry} = \boldsymbol{0}$ which leaves only the last term of eq. (8). Setting $\widehat{\boldsymbol{W}_{key}} \in \mathbb{R}^{D_k \times D_p}$ to the identity matrix (with appropriate row padding), yields $\mathbf{A}_{q,k} = \boldsymbol{v}^\top \boldsymbol{r}_\delta$ where $\quad \boldsymbol{\delta} := \boldsymbol{k} - \boldsymbol{q}$. Above, we have assumed that $D_p \leq D_k$ such that no information from $\boldsymbol{r}_\delta$ is lost.

Now, suppose that we could write:

$$\mathbf{A}_{q,k} = -\alpha(\|\boldsymbol{\delta} - \boldsymbol{\Delta}\|^2 + c) \tag{14}$$

for some constant $c$. In the above expression, the maximum attention score over $\mathbf{A}_{q,:}$ is $-\alpha c$ and it is reached for $\mathbf{A}_{q,k}$ with $\boldsymbol{\delta} = \boldsymbol{\Delta}$. On the other hand, the $\alpha$ coefficient can be used to scale arbitrarily the difference between $\mathbf{A}_{q,\Delta}$ and the other attention scores.

In this way, for $\boldsymbol{\delta} = \boldsymbol{\Delta}$, we have

$$\lim_{\alpha \to \infty} \text{softmax}(\mathbf{A}_{q,:})_k = \lim_{\alpha \to \infty} \frac{e^{-\alpha(\|\boldsymbol{\delta} - \boldsymbol{\Delta}\|^2 + c)}}{\sum_{k'} e^{-\alpha(\|(k-q')-\boldsymbol{\Delta}\|^2 + c)}}$$

$$= \lim_{\alpha \to \infty} \frac{e^{-\alpha\|\boldsymbol{\delta} - \boldsymbol{\Delta}\|^2}}{\sum_{k'} e^{-\alpha\|(k-q')-\boldsymbol{\Delta}\|^2}} = \frac{1}{1 + \lim_{\alpha \to \infty} \sum_{k' \neq k} e^{-\alpha\|(k-q')-\boldsymbol{\Delta}\|^2}} = 1$$

and for $\boldsymbol{\delta} \neq \boldsymbol{\Delta}$, the equation becomes $\lim_{\alpha \to \infty} \text{softmax}(\mathbf{A}_{q,:})_k = 0$, exactly as needed to satisfy the lemma statement.

What remains is to prove that there exist $\boldsymbol{v}$ and $\{\boldsymbol{r}_\delta\}_{\delta \in \mathbb{Z}^2}$ for which eq. (14) holds. Expanding the RHS of the equation, we have $-\alpha(\|\boldsymbol{\delta} - \boldsymbol{\Delta}\|^2 + c) = -\alpha(\|\boldsymbol{\delta}\|^2 + \|\boldsymbol{\Delta}\|^2 - 2\langle \boldsymbol{\delta}, \boldsymbol{\Delta} \rangle + c)$. Now if we set $\boldsymbol{v} = -\alpha(1, -2\boldsymbol{\Delta}_1, -2\boldsymbol{\Delta}_2)$ and $\boldsymbol{r}_\delta = (\|\boldsymbol{\delta}\|^2, \boldsymbol{\delta}_1, \boldsymbol{\delta}_2)$, then

$$\mathbf{A}_{q,k} = \boldsymbol{v}^\top \boldsymbol{r}_\delta = -\alpha(\|\boldsymbol{\delta}\|^2 - 2\boldsymbol{\Delta}_1 \boldsymbol{\delta}_1 - 2\boldsymbol{\Delta}_2 \boldsymbol{\delta}_2) = -\alpha(\|\boldsymbol{\delta}\|^2 - 2\langle \boldsymbol{\delta}, \boldsymbol{\Delta} \rangle) = -\alpha(\|\boldsymbol{\delta} - \boldsymbol{\Delta}\|^2 - \|\boldsymbol{\Delta}\|^2),$$

which matches eq. (14) with $c = -\|\boldsymbol{\Delta}\|^2$ and the proof is concluded. $\qquad \square$

**Remark on the magnitude of $\alpha$.** The exact representation of one pixel requires $\alpha$ (or the matrices $\boldsymbol{W}_{qry}$ and $\boldsymbol{W}_{key}$) to be arbitrary large, despite the fact that the attention probabilities of all other pixels converge exponentially to 0 as $\alpha$ grows. Nevertheless, practical implementations always rely on finite precision arithmetic for which a constant $\alpha$ suffices to satisfy our construction. For instance, since the smallest positive `float32` scalar is approximately $10^{-45}$, setting $\alpha = 46$ would suffice to obtain hard attention.

## 4 EXPERIMENTS

The aim of this section is to validate the applicability of our theoretical results—which state that self-attention *can* perform convolution—and to examine whether self-attention layers in practice do actually learn to operate like convolutional layers when trained on standard image classification tasks. In particular, we study the relationship between self-attention and convolution with *quadratic* and *learned* relative positional encodings. We find that, for both cases, the attention probabilities learned tend to respect the conditions of Lemma 1, supporting our hypothesis.

### 4.1 IMPLEMENTATION DETAILS

We study a fully attentional model consisting of six multi-head self-attention layers. As it has already been shown by Bello et al. (2019) that combining attention features with convolutional features improves performance on Cifar-100 and ImageNet, we do not focus on attaining state-of-the-art performance. Nevertheless, to validate that our model learns a meaningful classifier, we compare it to the standard ResNet18 (He et al., 2015) on the CIFAR-10 dataset (Krizhevsky et al.). In all experiments, we use a $2 \times 2$ invertible down-sampling (Jacobsen et al., 2018) on the input to reduce the size of the image. As the size of the attention coefficient tensors (stored during forward) scales quadratically with the size of the input image, *full* attention cannot be applied to bigger images. The fixed size representation of the input image is computed as the average pooling of the last layer representations and given to a linear classifier.

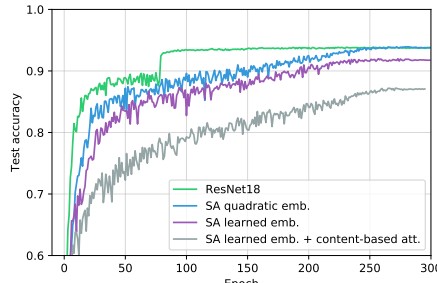

Figure 2: Test accuracy on CIFAR-10.

| Models | accuracy | # of params | # of FLOPS |
|---|---|---|---|
| ResNet18 | 0.938 | 11.2M | 1.1B |
| SA quadratic emb. | 0.938 | 12.1M | 6.2B |
| SA learned emb. | 0.918 | 12.3M | 6.2B |
| SA learned emb. + content | 0.871 | 29.5M | 15B |

Table 1: Test accuracy on CIFAR-10 and model sizes. SA stands for Self-Attention.

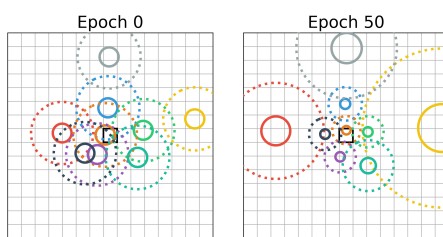
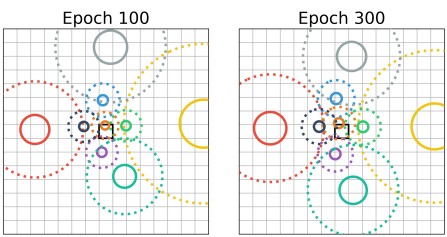

Figure 3: Centers of attention of each attention head (different colors) at layer 4 during the training with quadratic relative positional encoding. The central black square is the query pixel, whereas solid and dotted circles represent the 50% and 90% percentiles of each Gaussian, respectively.

We used the PyTorch library (Paszke et al., 2017) and based our implementation on PyTorch Transformers[5]. We release our code on Github[6] and hyper-parameters are listed in Table 2 (Appendix).

**Remark on accuracy.** To verify that our self-attention models perform reasonably well, we display in Figure 6 the evolution of the test accuracy on CIFAR-10 over the 300 epochs of training for our self-attention models against a small ResNet (Table 1). The ResNet is faster to converge, but we cannot ascertain whether this corresponds to an inherent property of the architecture or an artifact of the adopted optimization procedures. Our implementation could be optimized to exploit the locality of Gaussian attention probabilities and reduce significantly the number of FLOPS. We observed that learned embeddings with content-based attention were harder to train probably due to their increased number of parameters. We believe that the performance gap can be bridged to match the ResNet performance, but this is not the focus of this work.

## 4.2 QUADRATIC ENCODING

As a first step, we aim to verify that, with the relative position encoding introduced in equation (9), attention layers learn to behave like convolutional layers. We train nine attention heads at each layer to be on par with the $3 \times 3$ kernels used predominantly by the ResNet architecture. The center of attention of each head $h$ is initialized to $\mathbf{\Delta}^{(h)} \sim \mathcal{N}(\mathbf{0}, 2\mathbf{I}_2)$.

Figure 3 shows how the initial positions of the heads (different colors) at layer 4 changed during training. We can see that after optimization, the heads attend on specific pixel of the image forming a grid around the query pixel. Our intuition that Self-Attention applied to images learns convolutional filters around the queried pixel is confirmed.

Figure 4 displays all attention head at each layer of the model at the end of the training. It can be seen that in the first few layers the heads tend to focus on local patterns (layers 1 and 2), while deeper layers (layers 3-6) also attend to larger patterns by positioning the center of attention further from the queried pixel position. We also include in the Appendix a plot of the attention positions for a higher number of heads ($N_h = 16$). Figure 14 displays both local patterns similar to CNN and long range dependencies. Interestingly, attention heads do not overlap and seem to take an arrangement maximizing the coverage of the input space.

---

[5]github.com/huggingface/pytorch-transformers
[6]github.com/epfml/attention-cnn

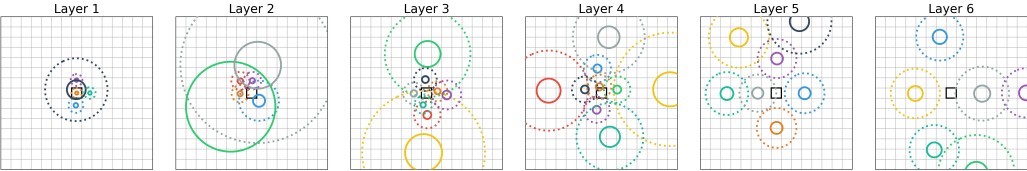

Figure 4: Centers of attention of each attention head (different colors) for the 6 self-attention layers using quadratic positional encoding. The central black square is the query pixel, whereas solid and dotted circles represent the 50% and 90% percentiles of each Gaussian, respectively.

### 4.3 LEARNED RELATIVE POSITIONAL ENCODING

We move on to study the positional encoding used in practice by fully-attentional models on images.

We implemented the 2D relative positional encoding scheme used by (Ramachandran et al., 2019; Bello et al., 2019): we learn a $\lfloor D_p/2 \rfloor$ position encoding vector for each row and each column pixel shift. Hence, the relative positional encoding of a key pixel at position $k$ with a query pixel at position $q$ is the concatenation of the row shift embedding $\delta_1$ and the column shift embedding $\delta_2$ (where $\delta = k - q$). We chose $D_p = D_{out} = 400$ in the experiment. We differ from their (unpublished) implementation in the following points: (*i*) we do not use convolution stem and ResNet bottlenecks for downsampling, but only a $2 \times 2$ invertible downsampling layer (Jacobsen et al., 2018) at input, (*ii*) we use $D_h = D_{out}$ instead of $D_h = D_{out}/N_h$ backed by our theory that the effective number of learned filters is $\min(D_h, D_{out})$.

At first, we discard the input data and compute the attention scores solely as the last term of eq. (8). The attention probabilities of each head at each layer are displayed on Figure 5. The figure confirms our hypothesis for the first two layers and partially for the third: even when left to learn the positional encoding scheme from randomly initialized vectors, certain self-attention heads (depicted on the left) learn to attend to individual pixels, closely matching the condition of Lemma 1 and thus Theorem 1. At the same time, other heads pay attention to horizontally-symmetric but non-localized patterns, as well as to long-range pixel inter-dependencies.

We move on to a more realistic setting where the attention scores are computed using both positional and content-based attention (i.e., $q^\top k + q^\top r$ in (Ramachandran et al., 2019)) which corresponds to a full-blown standalone self-attention model.

The attention probabilities of each head at each layer are displayed in Figure 6. We average the attention probabilities over a batch of 100 test images to outline the focus of each head and remove the dependency on the input image. Our hypothesis is confirmed for some heads of layer 2 and 3: even when left to learn the encoding from the data, certain self-attention heads only exploit position-based attention to attend to distinct pixels at a fixed shift from the query pixel reproducing the receptive field of a convolutional kernel. Other heads use more content-based attention (see Figures 8 to 10 in Appendix for non-averaged probabilities) leveraging the advantage of Self-Attention over CNN which does not contradict our theory. In practice, it was shown by Bello et al. (2019) that combining CNN and self-attention features outperforms each taken separately. Our experiments shows that such combination is learned when optimizing an unconstrained fully-attentional model.

The similarity between convolution and multi-head self-attention is striking when the query pixel is slid over the image: the localized attention patterns visible in Figure 6 follow the query pixel. This characteristic behavior materializes when comparing Figure 6 with the attention probabilities at a different query pixel (see Figure 7 in Appendix). Attention patterns in layers 2 and 3 are not only localized but stand at a constant shift from the query pixel, similarly to convolving the receptive field of a convolutional kernel over an image. This phenomenon is made evident on our interactive website[7]. This tool is designed to explore different components of attention for diverse images with or without content-based attention. We believe that it is a useful instrument to further understand how MHSA learns to process images.

---

[7] `epfml.github.io/attention-cnn`

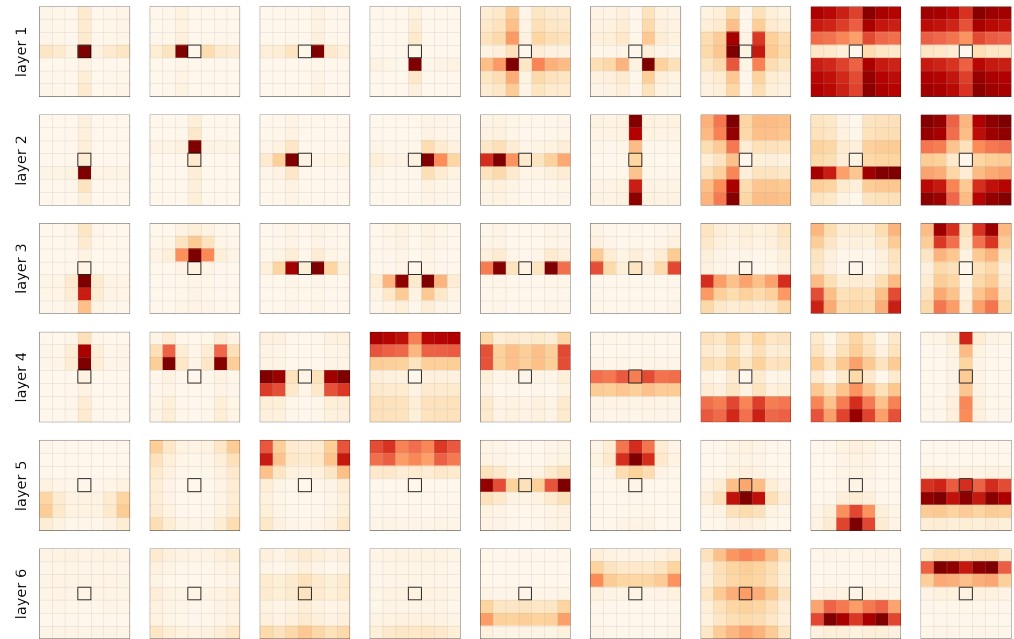

Figure 5: Attention probabilities of each head (*column*) at each layer (*row*) using learned relative positional encoding without content-based attention. The central black square is the query pixel. We reordered the heads for visualization and zoomed on the 7x7 pixels around the query pixel.

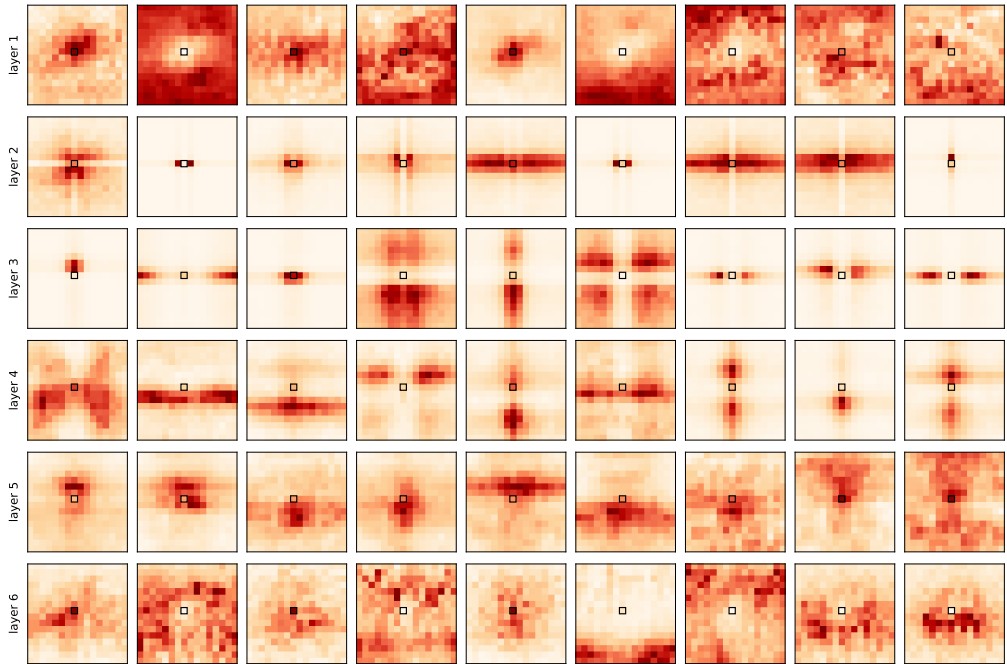

Figure 6: Attention probabilities for a model with 6 layers (*rows*) and 9 heads (*columns*) using learned relative positional encoding and content-content based attention. Attention maps are averaged over 100 test images to display head behavior and remove the dependence on the input content. The black square is the query pixel. More examples are presented in Appendix A.

## 5 RELATED WORK

In this section, we review the known differences and similarities between CNNs and transformers.

The use of CNN networks for text—at word level (Gehring et al., 2017) or character level (Kim, 2014)—is more seldom than transformers (or RNN). Transformers and convolutional models have been extensively compared empirically on tasks of Natural Language Processing and Neural Machine Translation. It was observed that transformers have a competitive advantage over convolutional model applied to text (Vaswani et al., 2017). It is only recently that Bello et al. (2019); Ramachandran et al. (2019) used transformers on images and showed that they achieve similar accuracy as ResNets. However, their comparison only covers performance and number of parameters and FLOPS but not expressive power.

Beyond performance and computational-cost comparisons of transformers and CNN, the study of expressiveness of these architectures has focused on their ability to capture long-term dependencies (Dai et al., 2019). Another interesting line of research has demonstrated that transformers are Turing-complete (Dehghani et al., 2018; Pérez et al., 2019), which is an important theoretical result but is not informative for practitioners. To the best of our knowledge, we are the first to show that the class of functions expressed by a layer of self-attention encloses all convolutional filters.

The closest work in bridging the gap between attention and convolution is due to Andreoli (2019). They cast attention and convolution into a unified framework leveraging tensor outer-product. In this framework, the receptive field of a convolution is represented by a "basis" tensor $\mathbf{A} \in \mathbb{R}^{K \times K \times H \times W \times H \times W}$. For instance, the receptive field of a classical $K \times K$ convolutional kernel would be encoded by $\mathbf{A}_{\boldsymbol{\Delta}, \boldsymbol{q}, \boldsymbol{k}} = \mathbb{1}\{\boldsymbol{k} - \boldsymbol{q} = \boldsymbol{\Delta}\}$ for $\boldsymbol{\Delta} \in \mathbb{\Delta}_K$. The author distinguishes this *index-based* convolution with *content-based* convolution where $\mathbf{A}$ is computed from the value of the input, e.g., using a key/query dot-product attention. Our work moves further and presents sufficient conditions for relative positional encoding injected into the input content (as done in practice) to allow *content-based* convolution to express any *index-based* convolution. We further show experimentally that such behavior is learned in practice.

## 6 CONCLUSION

We showed that self-attention layers applied to images can express any convolutional layer (given sufficiently many heads) and that fully-attentional models learn to combine local behavior (similar to convolution) and global attention based on input content. More generally, fully-attentional models seem to learn a generalization of CNNs where the kernel pattern is learned at the same time as the filters—similar to deformable convolutions (Dai et al., 2017; Zampieri, 2019). Interesting directions for future work include translating existing insights from the rich CNNs literature back to transformers on various data modalities, including images, text and time series.

ACKNOWLEDGMENTS

Jean-Baptiste Cordonnier is thankful to the Swiss Data Science Center (SDSC) for funding this work. Andreas Loukas was supported by the Swiss National Science Foundation (project "Deep Learning for Graph Structured Data", grant number PZ00P2 179981).

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

# APPENDIX

## A  MORE EXAMPLES WITH CONTENT-BASED ATTENTION

We present more examples of attention probabilities computed by self-attention model. Figure 7 shows average attention at a different query pixel than Figure 6. Figures 8 to 10 display attention for single images.

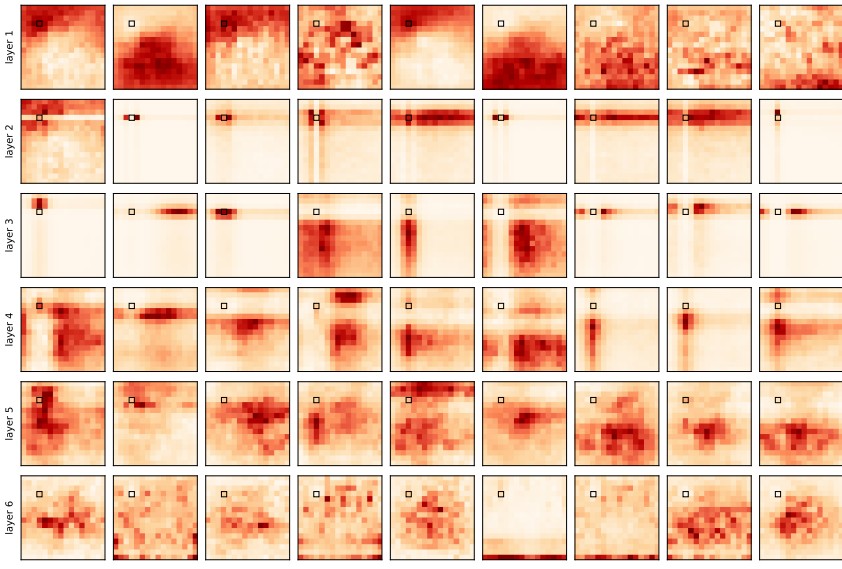

Figure 7: Attention probabilities for a model with 6 layers (*rows*) and 9 heads (*columns*) using learned relative positional encoding and content-content attention. We present the average of 100 test images. The black square is the query pixel.

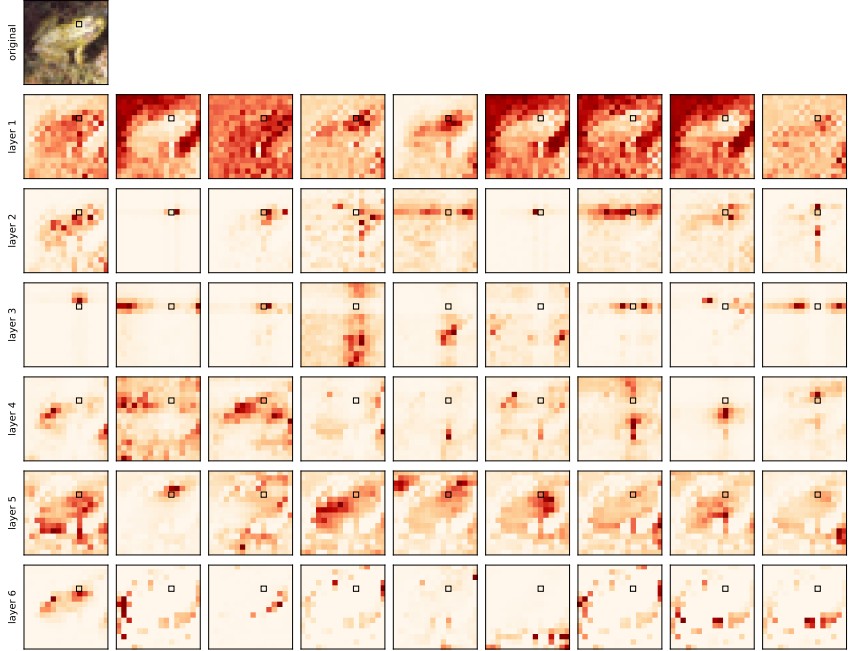

Figure 8: Attention probabilities for a model with 6 layers (*rows*) and 9 heads (*columns*) using learned relative positional encoding and content-content based attention. The query pixel (black square) is on the frog head.

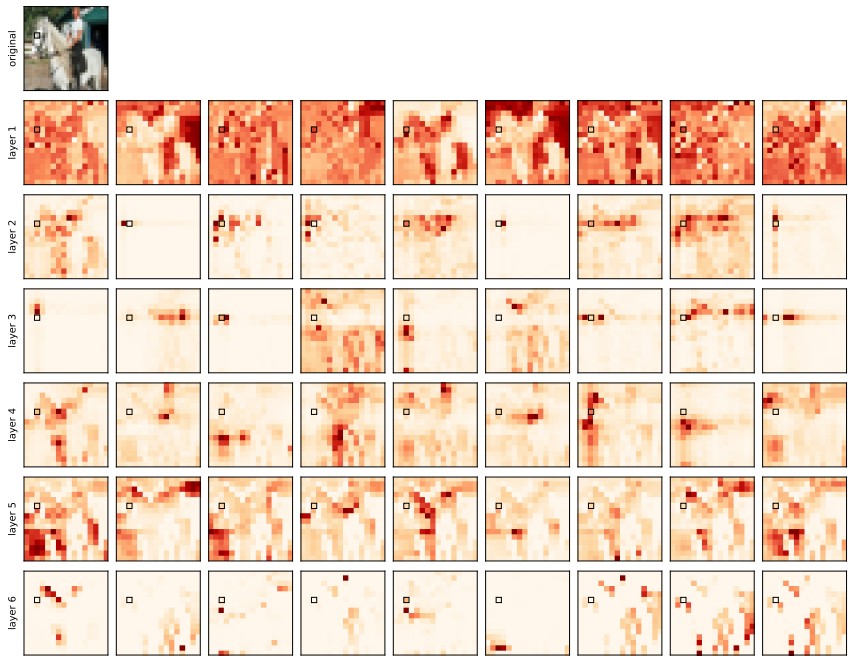

Figure 9: Attention probabilities for a model with 6 layers (*rows*) and 9 heads (*columns*) using learned relative positional encoding and content-content based attention. The query pixel (black square) is on the horse head.

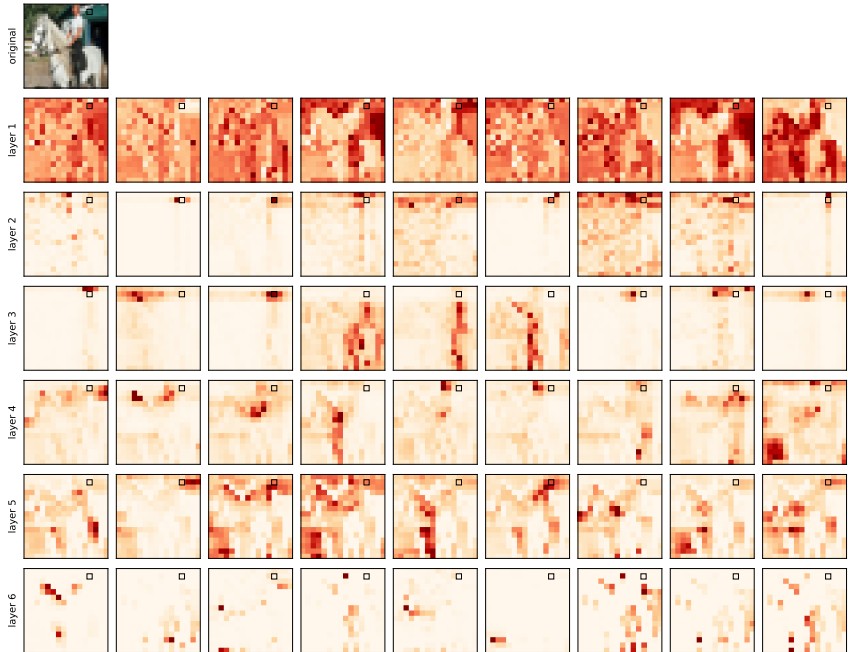

Figure 10: Attention probabilities for a model with 6 layers (*rows*) and 9 heads (*columns*) using learned relative positional encoding and content-content based attention. The query pixel (black square) is on the building in the background.

## B  Hyper-parameters used in our Experiments

| Hyper-parameters | |
|---|---|
| number of layers | 6 |
| number of heads | 9 |
| hidden dimension | 400 |
| intermediate dimension | 512 |
| invertible pooling width | 2 |
| dropout probability | 0.1 |
| layer normalization epsilon | $10^{-12}$ |
| number of epochs | 300 |
| batch size | 100 |
| learning rate | 0.1 |
| weight decay | 0.0001 |
| momentum | 0.9 |
| cosine decay | ✓ |
| linear warm up ratio | 0.05 |

Table 2: Self-attention network parameters

## C  Positional Encoding References

| Model | type of positional encoding | | | relative |
|---|---|---|---|---|
| | sinusoids | learned | quadratic | |
| Vaswani et al. (2017) | ✓ | | | |
| Radford et al. (2018) | | ✓ | | |
| Devlin et al. (2018) | | ✓ | | |
| Dai et al. (2019) | ✓ | | | ✓ |
| Yang et al. (2019) | ✓ | | | ✓ |
| Bello et al. (2019) | | ✓ | | ✓ |
| Ramachandran et al. (2019) | | ✓ | | ✓ |
| Our work | | ✓ | ✓ | ✓ |

Table 3: Types of positional encoding used by transformers models applied to text (*top*) and images (*bottom*). When multiple encoding types have been tried, we report the one advised by the authors.

## D  Generalized Lemma 1

We present a generalization of Lemma 1 that replaces the necessity of hard attention (to single pixels) by a milder assumption: the attention probabilities should span the grid receptive field. The conditions of this Lemma are still satisfied by Lemma 2, hence Theorem 1 follows.

**Lemma 3.** *Consider a multi-head self-attention layer consisting of $N_h \geq K^2$ heads, $D_h \geq D_{out}$ and let $\omega : [H] \times [W] \rightarrow [HW]$ be a pixel indexing. Then, for any convolutional layer with a $K \times K$ kernel and $D_{out}$ output channels, there exists $\{W_{val}^{(h)}\}_{h \in [N_h]}$ and $W_{out}$ such that $\mathrm{MHSA}(\mathbf{X}) = \mathrm{Conv}(\mathbf{X})$ for every $\mathbf{X} \in \mathbb{R}^{W \times H \times D_{in}}$ if and only if, for all $q \in [H] \times [W]$,* [8]

$$\mathrm{span}(\{e_{\omega(q+\Delta)} \in \mathbb{R}^{HW} : \Delta \in \mathbb{A}_K\}) \subseteq \mathrm{span}(\{\mathrm{vect}(\mathrm{softmax}(\mathbf{A}_{q,:}^{(h)})) : h \in [N_h]\}).$$

---

[8] the vectorization operator $\mathrm{vect}(\cdot)$ flattens a matrix into a vector

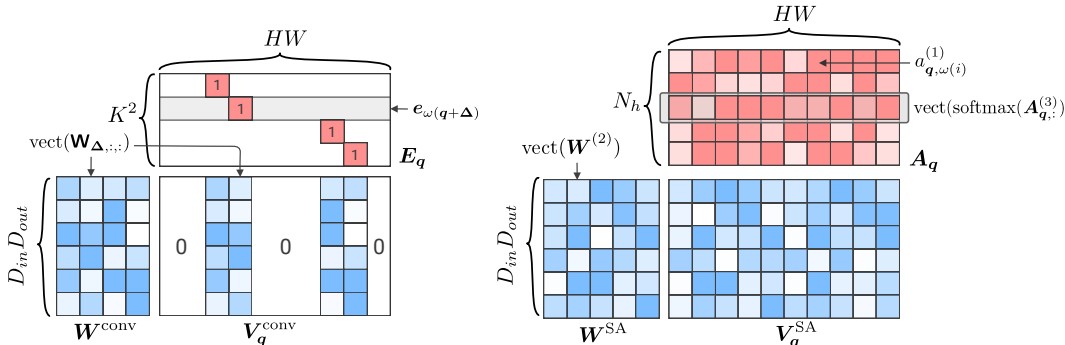

Figure 11: Factorization of the vectorized weight matrices $\boldsymbol{V}_q^{\text{conv}}$ and $\boldsymbol{V}_q^{\text{SA}}$ used to compute the output at position $\boldsymbol{q}$ for an input image of dimension $H \times W$. On the *left*: a convolution of kernel $2 \times 2$, on the *right*: a self-attention with $N_h = 5$ heads. $D_{in} = 2$, $D_{out} = 3$ in both cases.

*Proof.* Our first step will be to rework the expression of the Multi-Head Self-Attention operator from equation (1) and equation (4) such that the effect of the multiple heads becomes more transparent:

$$\text{MHSA}(\mathbf{X}) = \boldsymbol{b}_{out} + \sum_{h \in [N_h]} \text{softmax}(\mathbf{A}^{(h)})\mathbf{X} \underbrace{\boldsymbol{W}_{val}^{(h)} \boldsymbol{W}_{out}[(h-1)D_h + 1 : hD_h + 1]}_{\boldsymbol{W}^{(h)}} \tag{15}$$

Note that each head's value matrix $\boldsymbol{W}_{val}^{(h)} \in \mathbb{R}^{D_{in} \times D_h}$ and each block of the projection matrix $\boldsymbol{W}_{out}$ of dimension $D_h \times D_{out}$ are learned. Assuming that $D_h \geq D_{out}$, we can replace each pair of matrices by a learned matrix $\boldsymbol{W}^{(h)}$ for each head. We consider one output pixel of the multi-head self-attention and drop the bias term for simplicity:

$$\text{MHSA}(\mathbf{X})_{\boldsymbol{q},:} = \sum_{h \in [N_h]} \Big( \sum_{\boldsymbol{k}} a_{\boldsymbol{q},\boldsymbol{k}}^{(h)} \mathbf{X}_{\boldsymbol{k},:} \Big) \boldsymbol{W}^{(h)} = \sum_{\boldsymbol{k}} \mathbf{X}_{\boldsymbol{k},:} \Big( \underbrace{\sum_{h \in [N_h]} a_{\boldsymbol{q},\boldsymbol{k}}^{(h)} \boldsymbol{W}^{(h)}}_{\boldsymbol{W}_{\boldsymbol{q},\boldsymbol{k}}^{\text{SA}} \in \mathbb{R}^{D_{in} \times D_{out}}} \Big), \tag{16}$$

with $a_{\boldsymbol{q},\boldsymbol{k}}^{(h)} = \text{softmax}(\mathbf{A}_{\boldsymbol{q},:}^{(h)})_{\boldsymbol{k}}$. We rewrite the output of a convolution at pixel $\boldsymbol{q}$ in the same manner:

$$\text{Conv}(\mathbf{X})_{\boldsymbol{q},:} = \sum_{\boldsymbol{\Delta} \in \mathbb{\Delta}_K} \mathbf{X}_{\boldsymbol{q}+\boldsymbol{\Delta},:} \mathbf{W}_{\boldsymbol{\Delta},:,:} = \sum_{\boldsymbol{k} \in [H] \times [W]} \mathbf{X}_{\boldsymbol{k},:} \underbrace{\mathbb{1}_{\{\boldsymbol{k}-\boldsymbol{q} \in \mathbb{\Delta}_K\}} \mathbf{W}_{\boldsymbol{k}-\boldsymbol{q},:,:}}_{\boldsymbol{W}_{\boldsymbol{q},\boldsymbol{k}}^{\text{conv}} \in \mathbb{R}^{D_{in} \times D_{out}}} . \tag{17}$$

Equality between equations (16) and (17) holds for any input $\mathbf{X}$ if and only if the linear transformations for each pair of key/query pixels are equal, i.e. $\boldsymbol{W}_{\boldsymbol{q},\boldsymbol{k}}^{\text{conv}} = \boldsymbol{W}_{\boldsymbol{q},\boldsymbol{k}}^{\text{SA}} \ \forall \boldsymbol{q}, \boldsymbol{k}$. We vectorize the weight matrices into matrices of dimension $D_{in} D_{out} \times HW$ as $\boldsymbol{V}_q^{\text{conv}} := [\text{vect}(\boldsymbol{W}_{\boldsymbol{q},\boldsymbol{k}}^{\text{conv}})]_{\boldsymbol{k} \in [H] \times [W]}$ and $\boldsymbol{V}_q^{\text{SA}} := [\text{vect}(\boldsymbol{W}_{\boldsymbol{q},\boldsymbol{k}}^{\text{SA}})]_{\boldsymbol{k} \in [H] \times [W]}$. Hence, to show that $\text{Conv}(\mathbf{X}) = \text{MHSA}(\mathbf{X})$ for all $\mathbf{X}$, we must show that $\boldsymbol{V}_q^{\text{conv}} = \boldsymbol{V}_q^{\text{SA}}$ for all $\boldsymbol{q}$.

The matrix $\boldsymbol{V}_q^{\text{conv}}$ has a restricted support: only the columns associated with a pixel shift $\boldsymbol{\Delta} \in \mathbb{\Delta}_K$ in the receptive field of pixel $\boldsymbol{q}$ can be non-zero. This leads to the factorization $\boldsymbol{V}_q^{\text{conv}} = \boldsymbol{W}^{\text{conv}} \boldsymbol{E}_q$ displayed in Figure 11 where $\boldsymbol{W}^{\text{conv}} \in \mathbb{R}^{D_{in} D_{out} \times K^2}$ and $\boldsymbol{E}_q \in \mathbb{R}^{K^2 \times HW}$. Given an ordering of the shifts $\boldsymbol{\Delta} \in \mathbb{\Delta}_K$ indexed by $j$, set $(\boldsymbol{W}^{\text{conv}})_{:,j} = \text{vect}(\mathbf{W}_{\boldsymbol{\Delta},:,:})$ and $(\boldsymbol{E}_q)_{j,:} = \boldsymbol{e}_{\omega(\boldsymbol{q}+\boldsymbol{\Delta})}$. On the other hand, we decompose $\boldsymbol{V}_q^{\text{SA}} = \boldsymbol{W}^{\text{SA}} \boldsymbol{A}_q$ with $(\boldsymbol{W}^{\text{SA}})_{:,h} = \text{vect}(\boldsymbol{W}^{(h)})$ and $(\boldsymbol{A}_q)_{h,i} = a_{\boldsymbol{q},\omega(i)}^{(h)}$.

The proof is concluded by showing that $\text{row}(\boldsymbol{E}_q) \subseteq \text{row}(\boldsymbol{A}_q)$ is a necessary and sufficient condition for the existence of a $\boldsymbol{W}^{\text{SA}}$ such that any $\boldsymbol{V}_q^{\text{conv}} = \boldsymbol{W}^{\text{conv}} \boldsymbol{E}_q$ can be written as $\boldsymbol{W}^{\text{SA}} \boldsymbol{A}_q$.

**Sufficient.** Given that $\text{row}(\boldsymbol{E}_q) \subseteq \text{row}(\boldsymbol{A}_q)$, there exists $\Phi \in \mathbb{R}^{K^2 \times N_h}$ such that $\boldsymbol{E}_q = \Phi \boldsymbol{A}_q$ and a valid decomposition is $\boldsymbol{W}^{\text{SA}} = \boldsymbol{W}^{\text{conv}} \Phi$ which gives $\boldsymbol{W}^{\text{SA}} \boldsymbol{A}_q = \boldsymbol{V}_q^{\text{conv}}$.

**Necessary.** Assume there exists $\boldsymbol{x} \in \mathbb{R}^{HW}$ such that $\boldsymbol{x} \in \text{row}(\boldsymbol{E}_q)$ and $\boldsymbol{x} \notin \text{row}(\boldsymbol{A}_q)$ and set $\boldsymbol{x}^\top$ to be a row of $\boldsymbol{V}_q^{\text{conv}}$. Then, $\boldsymbol{W}^{\text{SA}} \boldsymbol{A}_q \neq \boldsymbol{V}_q^{\text{conv}}$ for any $\boldsymbol{W}^{\text{SA}}$ and there is no possible decomposition.

$\square$

# E   GENERALIZED QUADRATIC POSITIONAL ENCODING

We noticed the similarity of the attention probabilities in the quadratic positional encoding (Section 3) to isotropic bivariate Gaussian distributions with bounded support:

$$\text{softmax}(\mathbf{A}_{q,:})_{\boldsymbol{k}} = \frac{e^{-\alpha\|(\boldsymbol{k}-\boldsymbol{q})-\boldsymbol{\Delta}\|^2}}{\sum_{\boldsymbol{k}'\in[W]\times[H]} e^{-\alpha\|(\boldsymbol{k}'-\boldsymbol{q})-\boldsymbol{\Delta}\|^2}}\,. \tag{18}$$

Building on this observation, we further extended our attention mechanism to non-isotropic Gaussian distribution over pixel positions. Each head is parametrized by a center of attention $\boldsymbol{\Delta}$ and a covariance matrix $\boldsymbol{\Sigma}$ to obtain the following attention scores,

$$\mathbf{A}_{q,k} = -\frac{1}{2}(\boldsymbol{\delta}-\boldsymbol{\Delta})^\top\boldsymbol{\Sigma}^{-1}(\boldsymbol{\delta}-\boldsymbol{\Delta}) = -\frac{1}{2}\boldsymbol{\delta}^\top\boldsymbol{\Sigma}^{-1}\boldsymbol{\delta} + \boldsymbol{\delta}^\top\boldsymbol{\Sigma}^{-1}\boldsymbol{\Delta} - \frac{1}{2}\boldsymbol{\Delta}^\top\boldsymbol{\Sigma}^{-1}\boldsymbol{\Delta}\,, \tag{19}$$

where, once more, $\boldsymbol{\delta} = \boldsymbol{k} - \boldsymbol{q}$. The last term can be discarded because the softmax is shift invariant and we rewrite the attention coefficient as a dot product between the head target vector $\boldsymbol{v}$ and the relative position encoding $\boldsymbol{r_\delta}$ (consisting of the first and second order combinations of the shift in pixels $\boldsymbol{\delta}$):

$$\boldsymbol{v} = \frac{1}{2}(2(\boldsymbol{\Sigma}^{-1}\boldsymbol{\Delta})_1, 2(\boldsymbol{\Sigma}^{-1}\boldsymbol{\Delta})_2, -\boldsymbol{\Sigma}^{-1}_{1,1}, -\boldsymbol{\Sigma}^{-1}_{2,2}, -2\cdot\boldsymbol{\Sigma}^{-1}_{1,2})^\top \text{ and } \boldsymbol{r_\delta} = (\boldsymbol{\delta}_1, \boldsymbol{\delta}_2, \boldsymbol{\delta}_1^2, \boldsymbol{\delta}_2^2, \boldsymbol{\delta}_1\boldsymbol{\delta}_2)^\top\,.$$

**Evaluation.**   We trained our model using this generalized quadratic relative position encoding. We were curious to see if, using the above encoding the self-attention model would learn to attend to non-isotropic groups of pixels—thus forming unseen patterns in CNNs. Each head was parametrized by $\boldsymbol{\Delta}\in\mathbb{R}^2$ and $\boldsymbol{\Sigma}^{-1/2}\in\mathbb{R}^{2\times 2}$ to ensure that the covariance matrix remained positive semi-definite. We initialized the center of attention to $\boldsymbol{\Delta}^{(h)}\sim\mathcal{N}(\mathbf{0}, 2\boldsymbol{I}_2)$ and $\boldsymbol{\Sigma}^{-1/2} = \boldsymbol{I}_2 + \mathcal{N}(\mathbf{0}, 0.01\boldsymbol{I}_2)$ so that initial attention probabilities were close to an isotropic Gaussian. Figure 12 shows that the network did learn non-isotropic attention probability patterns, especially in high layers. Nevertheless, the fact that we do not obtain any performance improvement seems to suggest that attention non-isotropy is not particularly helpful in practice—the quadratic positional encoding suffices.

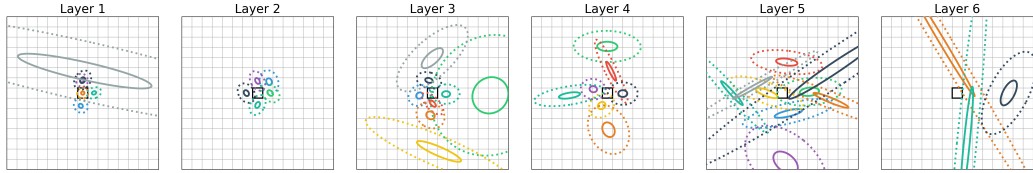

Figure 12: Centers of attention of each attention head (different colors) for the 6 self-attention layers using non-isotropic Gaussian parametrization. The central black square is the query pixel, whereas solid and dotted circles represent the 50% and 90% percentiles of each Gaussian, respectively.

**Pruning degenerated heads.**   Some non-isotropic attention heads attend on "non-intuitive" patches of pixels: either attending a very thin stripe of pixels, when $\boldsymbol{\Sigma}^{-1}$ was almost singular, or attending all pixels uniformly, when $\boldsymbol{\Sigma}^{-1}$ was close to $\mathbf{0}$ (i.e. constant attention scores). We asked ourselves, are such attention patterns indeed useful for the model or are these heads degenerated and unused? To find out, we pruned all heads having largest eigen-values smaller than $10^{-5}$ or condition number (ratio of the biggest and smallest eigen-values) greater than $10^5$. Specifically in our model with 6-layer and 9-heads each, we pruned $[2, 4, 1, 2, 6, 0]$ heads from the first to the last layer. This means that these layers cannot express a $3\times 3$ kernel anymore. As shown in yellow on fig. 2, this ablation initially hurts a bit the performance, probably due to off biases, but after a few epochs of continued training with a smaller learning rate (divided by 10) the accuracy recovers its unpruned value. Hence, without sacrificing performance, we reduce the size of the parameters and the number of FLOPS by a fourth.

# F   INCREASING THE NUMBER OF HEADS

For completeness, we also tested increasing the number of heads of our architecture from 9 to 16.

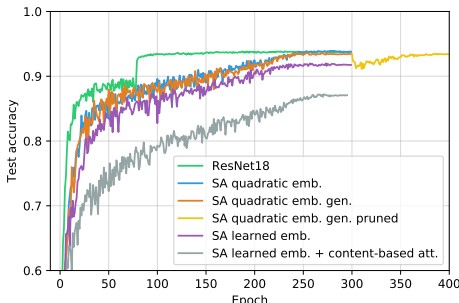

Figure 13: Evolution of test accuracy on CIFAR-10. Pruned model (*yellow*) is continued training of the non-isotropic model (*orange*).

| Models | accuracy | # of params | # of FLOPS |
|---|---|---|---|
| ResNet18 | 0.938 | 11.2M | 1.1B |
| SA quadratic emb. | 0.938 | 12.1M | 6.2B |
| SA quadratic emb. gen. | 0.934 | 12.1M | 6.2B |
| SA quadratic emb. gen. pruned | 0.934 | 9.7M | 4.9B |
| SA learned emb. | 0.918 | 12.3M | 6.2B |
| SA learned emb. + content | 0.871 | 29.5M | 15B |

Table 4: Number of parameters and accuracy on CIFAR-10 per model. SA stands for Self-Attention.

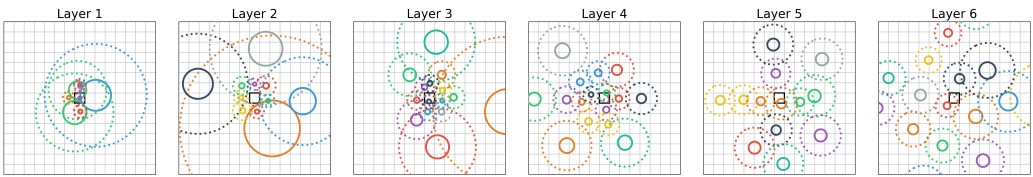

Figure 14: Centers of attention for 16 attention heads (different colors) for the 6 self-attention layers using quadratic positional encoding. The central black square is the query pixel, whereas solid and dotted circles represent the 50% and 90% percentiles of each Gaussian, respectively.

Similar to Figure 4, we see that the network distinguishes two main types of attention patterns. Localized heads (i.e., those that attend to nearly individual pixels) appear more frequently in the first few layers. The self-attention layer uses these heads to act in a manner similar to how convolutional layers do. Heads with less-localized attention become more common at higher layers.

