# OpenReview forum: "On the Relationship between Self-Attention and Convolutional Layers"
_ICLR.cc/2020/Conference — Accept (Poster)_

### Official Review · AnonReviewer1 · 2019-10-15
**Official Blind Review #1**

**Rating:** 6

**Review:**

The paper shows both theoretically and in practice that self-attention can learn to act as convolutions. The main intuition is that every attention head can learn to attend individually to a given relative offset around each pixel. Given enough heads (K**2) such a layer can imitate a convolution with kernel size (K,K). This leads to the conclusion that self-attention is at least as powerful as CNNs are. This fact has been acknowledged by (at least part of) the community for a while (following a similar intuition) but as far as I know has never been formalized. Hence, although incremental I consider this an important contribution. The derivation of quadratic relative encoding is a nice theoretical construction. Experiments show improvements over learned relative attention, however, experiments are merely conducted on Cifar.

Finally, even though the contributions are somewhat marginal and the experiments are not quite enough to establish the new relative attention mechanism as being superior, I like this paper and consider its contributions valuable. The message of the paper deserves a larger audience and I therefore lean to accept despite some shortcomings.

**Experience Assessment:**

I have published one or two papers in this area.

**Review Assessment: Checking Correctness Of Derivations And Theory:**

I carefully checked the derivations and theory.

**Review Assessment: Checking Correctness Of Experiments:**

I carefully checked the experiments.

**Review Assessment: Thoroughness In Paper Reading:**

I read the paper at least twice and used my best judgement in assessing the paper.

---

> ### Author Response · Authors · 2019-11-13
> **Answer to Official Blind Review #1**
>
> We thank the reviewer for their time assessing our work and their constructive feedback.
>
> As a minor correction to what you wrote, we would like to highlight that we do not claim quadratic positional encoding to be superior to what is used in practice. We conducted experiments with quadratic encoding out of curiosity to examine if this positional encoding--crafted solely for the proof of our main theorem--could actually be learned in practice. The fact that this encoding yields decent practical performance indicates that our proof by construction is not superficial.
>
> We agree that this formal result of expressivity is a first step towards better understanding the relationship between self-attention and convolution.

---

### Official Review · AnonReviewer3 · 2019-10-21
**Official Blind Review #3**

**Rating:** 6

**Review:**

The paper claims that 1. multi-head self-attention(MHSA) is at least as powerful as convolutions by showing that a CONV can be cast as a special case of MHSA and 2. that in practice, MHSA often mimic convolutional layers.

These claims are interesting and timely, given that there has been a fair amount of recent work that have explored the use of self-attention(SA) on image tasks, either by composing SA with convolutions or replacing convolutions altogether with self-attention (examples of each are referenced in the paper). So should these claims be true, they would give theoretical evidence that SA can completely replace convolutions.

However, I think that the claims are exaggerated and misleading.
1. The theory shows an arguably contrived link between self-attention and convolution. Theorem 1 says that a convolution can be seen as a special case of MHSA, and the constructive proof (that chooses SA parameters to derive a convolution) shows a correspondence between the output of each head of MHSA and a D_out by D_in linear transform applied to the D_in features of a single pixel, with attention weight given entirely to this pixel (i.e. hard attention). The derivation relies heavily on the use of a relative encoding scheme that sets W_qry=W_key = 0 (usually referred to as W^Q, W^K in the self-attention literature, the linear maps applied to the queries and keys) i.e. the attention weights do not depend on the key/query values, but only their relative positions. Moreover, the softmax temperature (an interpretation of 1/alpha) is set arbitrarily close to 0 to make the softmax saturate and attain hard-attention. With these two constraints, I am sceptical as to whether you can really say that you are implementing self-attention. In standard practice when MHSA is used, W^Q and W^K are never set to zero, and the scale of the logits for the self-attention weights are controlled by normalising them with sqrt(D_k) (or sqrt(D_k/N_h), depnding on how you choose to deal with multiple heads). Furthermore, the derivation only holds for when stride=1 and padding=“SAME”, such that the spatial dimensions of the input (H & W) remain unchanged. In fact the padding is not really dealt with in the derivation, and it is unclear whether the result can generalise to convolutions with stride > 1, making the claim “MHSA layer … is at least as powerful as any convolutional layer” problematic. Hence although I think the derivation is mathematically correct, I think that the link that the derivation makes between convolutions and MHSA is somewhat contrived and not a useful observation in practice. I expect MHSA with learned W_qry and W_key will behave differently to when they are set to 0, and it would be much more interesting/relevant to see how their behaviour compares with convolutions in this more realistic setting.

2. The heavy dependence of the experiments on the quadratic encoding, the aforementioned contrived form of MHSA that was used to derive the link between convolutions and MHSA, makes the results not very relevant and the claim that "MHSA often mimic convolutional layers" rather misleading. It could be more relevant if quadratic encoding can replace standard MHSA parametererisations with learned W_qry and W_key, but I’m not convinced that this is the case. Although Figure 4 suggests that this SA with quadratic encoding gives similar test performance to ResNet18, I think that CIFAR-10 classification is too simple a task to claim that quadratic encoding can replace standard SA with learned W_qry and W_key, and I think results can look very different for harder problems e.g. ImageNet, MSCOCO - explored in Ramachandran et al - made possible because they use local SA as opposed to full SA. Experiments on these problems would be much more interesting and relevant. Note that the experiments using the learned relative positional encoding have “attention scores (are) computed using only the relative positions of the pixels and not the data” (I’m guessing this means W_qry=W_key=0 again). Hence the qualitative similarities between MHSA and convolutions only hold for the rather restricted case where I get the impression that self-attention has been unrealistically constrained only to increase its chance of behaving similarly to convolutions. Also the comparison in Figure 4 and Table 1 is being used to support the claim that self-attention can be as “powerful” as convolutions, but I think this is misleading because both quadratic and learned SA uses full SA, where each pixel attends to all pixels - this means the time & memory complexity of the algorithm is O((HW)^2), whereas for convolutions it is O(HW). So the expressiveness of SA that matches convolutions for this particular problem comes at a significant cost, to the extent that for bigger problems (ImageNet, MSCOCO) full SA is not feasible due to its quadratic memory requirement, whereas convolutions don’t face this problem. I think this should be pointed out more explicitly in the text, and think the claim that “self-attention is at least as powerful as convolutions” should be replaced with a more moderate statement such as “self-attention defines a family of functions that contains convolutions (of stride 1)”

Summary: Although the writing of the paper is clear and the derivation is mathematically correct as far as I can see, the link between self-attention and convolutions in the paper are fairly contrived, hence the contribution of the paper to the field is not so significant in my honest opinion.

********************
I appreciate the authors' response, and understand that the maths suggests a single head of MHA (in the original form) cannot exactly emulate a general convolution. But empirically, the localised attention patterns do seem to suggest that each head can behave similarly to a restricted form of convolution, where similar weights are given to the receptive field (the local patch) in the neighbourhood each input pixel. Perhaps an analysis of what special case of convolution each head can emulate would be interesting, given the empirically observed similarities in the qualitative behaviour.

With the more justified nuance of the findings of the paper, and together with the authors' significant efforts to make the evaluation more relevant and thorough, I will increase my score to "weak accept".

**Experience Assessment:**

I have published one or two papers in this area.

**Review Assessment: Checking Correctness Of Derivations And Theory:**

I carefully checked the derivations and theory.

**Review Assessment: Checking Correctness Of Experiments:**

I carefully checked the experiments.

**Review Assessment: Thoroughness In Paper Reading:**

I read the paper thoroughly.

---

> ### Author Response · Authors · 2019-11-13
> **Answer to Official Blind Review #3**
>
> We thank the reviewer for their time assessing our work and their constructive feedback.
> We address your concerns about the theory and the experiments and we have updated the submitted paper accordingly.
>
>
>
> 1. About the theory
>
> Our theoretical claim is that Multi-Head Self-Attention is at least as expressive as convolution, which is compatible with setting $W_{qry}$ and $W_{key}$ to zero in the proof. As the input pixel positions of a convolutional kernel do not depend on the input image, setting attention scores based on the input data to 0 is coherent.
>
> Reviewer #2 and you share a common concern about setting the softmax temperature arbitrary close to 0 to attain hard-attention. We suggest you to consult our answer to their review and the remark we added in Section 3 about the scale of $\alpha$.
>
> Thank you for pointing out that we did not mention padding and stride. We considered the general convolution operator (defined in eq. (5)), however, the Conv2d layers implemented in deep learning frameworks also use stride, padding, dilation and padding_mode options. Following your review, we added a paragraph in Section 3 (page 4) to explain that our proof holds for "ZERO" padding and any stride by downsampling.
>
>
>
> 2. About the experiments
>
> We disagree that our experiments heavily depend on the quadratic encoding, which we only employ to illustrate our theory. Section 4.2 shows that the reparametrization we do in the proof is learnable (and performs reasonably well). This demonstrates that our argument for expressivity is not contrived (i.e., only works in theory), but it connects to what works in practice. At the same time, we would like to stress that we do not claim that the quadratic encoding can replace standard SA with learned $W_{qry}$ and $W_{key}$.
>
> We also would like to highlight that we do not claim that MHSA with quadratic encoding should replace CNN in practice, but only that it can learn to behave like CNNs. You are right that the time and memory complexity of full Self-Attention are indeed deceptively costly and we made it more precise in the text. To avoid misleading the reader, we removed the vague word "powerful" from the paper (two occurrences) to clarify that we meant expressive power.
>
> Concerning ImageNet and MSCOCO.
> As you mention, full attention is not feasible on larger images without leveraging local attention. The experiments on more challenging datasets conducted by Ramachandran et al. (2019) are impressive but serve a different purpose: showing that local MHSA achieves state-of-the-art performance with competitive number of parameters and number of FLOPS on classification/segmentation of large images. Local Attention would force the self-attention heads to attend only to local patches, hence defeating our goal to show that Self-Attention behaves like CNN by attending to neighboring pixels at fixed shifts.
>
> Concerning attention based on data (learned $W_{key}$ and $W_{qry}$).
> Disentangling position and content attention was a first step toward better understanding of how MHSA processes images in practical settings. To connect our findings with the full-blown MHSA in practice, we conducted more experiments with learned matrices $W_{key}$ and $W_{qry}$: Figure 6 (added to the paper) is the counterpart of Figure 5 when content-content attention is enabled, i.e. $q^\top k + q^\top r$ attention as in (Ramachandran et al., 2019). We averaged the attention probabilities over a batch of 100 test images to remove the dependence on the input content and observe if some heads probabilities are very localized to some pixels around the query pixels. The hypothesis that some attention heads focus on pixels at a fixed shift from the query pixel is confirmed. Other heads tend to use more content-based attention (see Figure 8-10 in Appendix for non-averaged probabilities) leveraging the advantage of Self-Attention over CNN (which does not contradict our theory). We will share (after deanonymization) an interactive website to visualize attention maps (with/without data content) for different images/batch/query pixels. In the meantime, reviewers can consider this demo GIF (animation of Figure 6 and 7) and appreciate the translation of the attended patches when sliding the query pixel (similar to sliding a convolutional kernel). https://drive.google.com/file/d/1METSetroUA2qd2slol9wt7YxucJslAmF/view?usp=sharing

---

> > ### Comment · AnonReviewer3 · 2019-11-13
> > **Reviewer's reply to the author rebuttal**
> >
> > "Following your review, we added a paragraph in Section 3 (page 4) to explain that our proof holds for "ZERO" padding and any stride by downsampling."
> > - Thanks, this makes sense.
> >
> > "We disagree that our experiments heavily depend on the quadratic encoding, which we only employ to illustrate our theory. Section 4.2 shows that the reparametrization we do in the proof is learnable (and performs reasonably well). This demonstrates that our argument for expressivity is not contrived (i.e., only works in theory), but it connects to what works in practice."
> > - I'd argue that even the 'learned relative positional encoding' results aren't very relevant since W_key=W_qry=0 there as well. Claiming that "learned fully-attentional models do behave similar to CNNs in practice" requires showing results on settings of learned fully-attentional models used in practice, i.e. learning W_key and W_qry (this leads to the next comment).
> >
> > "we conducted more experiments with learned matrices W_key and W_qry: Figure 6"
> > - Hence I think the addition of these experiments makes the paper much more interesting and relevant, but also think the analysis of the results and how it relates to the theory should be further clarified: the theory suggests that when W_key=W_qry=0 and fixed and alpha is sufficiently large, each attention head gives weight exclusively to a pixel, and the aggregation of information of these pixels is achieved by W_out that combines the output of each head. Figure 5 for W_key=W_qry=0 indeed shows sparse attention weights, somewhat in line with the theory. However Figure 6 where W_key and W_qry are learned, the attention weights are sparse only for a few heads in layers 2 and 3, and for others the pattern is fairly dense, at least very far from giving weight exclusively to a pixel. Hence here, the connection of the results to the theory is weak, and I disagree with your statement in the rebuttal that "The hypothesis that some attention heads focus on pixels at a fixed shift from the query pixel is confirmed." You've shown this for very few attention heads, and the rather differing behaviour of the rest suggest that the theory is not being shown to hold in practice.
> >
> > "Other heads tend to use more content-based attention (see Figure 8-10 in Appendix for non-averaged probabilities) leveraging the advantage of Self-Attention over CNN (which does not contradict our theory)."
> > - Indeed it doesn't contradict the theory, but it suggests that in practice self-attention doesn't behave like CNNs, at least not for the reasons that were given in the theory.
> >
> > "reviewers can consider this demo GIF (animation of Figure 6 and 7) and appreciate the translation of the attended patches when sliding the query pixel (similar to sliding a convolutional kernel)."
> > - these localised attention patterns that emerge when you slide the queries are interesting, and suggest in fact that the output of each head acts like a convolution, and NOT the aggregation of the heads (which the theory suggests). This was in fact what I was intuitively expecting when I first read the abstract of the paper, being more aligned with personal hands-on experience with self-attention, and I was disappointed to find out in the derivation that each head corresponds to a single pixel, a much more unrealistic scenario.
> >
> > So in summary, I think there is still some work that can be done regarding the link between the theory and the experiments, either by modifying the theory to better align with the experiments for the general, realistic case (learning W_qry and W_key) or weakening the claim that "learned fully-attentional models do behave similar to CNNs in practice" to "learned fully-attentional models can behave similar to CNNs".
> >
> > I also still think the analysis would be much more interesting and relevant had you explored the attention patterns that arise in fully-attentional models that are comparable with SOTA convolutional models, such as those in Ramachandran et al. (2019), applied to more realistic datasets such as ImageNet and MSCOCO. Even without any theory, it would at least help understand the behaviour of attention in these realistic models with a competitive edge.

---

> > > ### Author Response · Authors · 2019-11-15
> > > **Discussion with Reviewer #3**
> > >
> > > We would like to thank the reviewer for actively participating in the discussion.
> > >
> > >
> > >
> > > "these localised attention patterns that emerge when you slide the queries are interesting, and suggest in fact that the output of each head acts like a convolution, and NOT the aggregation of the heads (which the theory suggests)."
> > >
> > > This seems to be a key misunderstanding that we wish to clarify. A single head does not operate as a convolution, though it acts on one of the pixels/patches of the receptive field of the convolutional kernel. (Figures 6, 7 and the GIF display the attention probabilities used to compute the weighted average of the input, not the convolution kernel weights themselves.) To convince you, we present three arguments showing that a single head SA cannot perform general convolution:
> > >
> > > 1. To compute the output of a convolution at a given location, the $C_{in}$ input channels of the neighboring pixels can be linearly transformed independently by the different slices of the weight tensor of shape $K\times K \times C_{in} \times C_{out}$. This cannot be done by a single head of self-attention.
> > >
> > >     For example, a convolutional layer can transform an RGB image ($C_{in}=3$) into a grey-scale image ($C_{out}=1$) with a $(1,2)$ kernel by summing the red channel of each pixel with the green channel of the pixel at its right. A single-head self-attention layer cannot simulate such convolution because filter matrices applied at each shift position have to be the same (up to a positive rescaling allowed by the attention probabilities).
> > >
> > > 2. Due to the constraint imposed by the softmax, even for $C_{in} = C_{out} = 1$ a single head multi-attention cannot express a derivative kernel. Such kernels require positive and negative weights, which violates the constraint imposed on the attention probabilities.
> > >
> > > 3. Finally, we present a more formal argument why a single head self-attention (SA) layer cannot act like convolution. Let us momentarily consider an arbitrary convolution layer and suppose that
> > >
> > > $$
> > > Y_q^{(\text{conv})} = \sum_{k} X_{k,:} W_{q-k} \in \mathbb{R}^{D_{out}}
> > > $$
> > >
> > > is its output at some pixel $q$. Further, let
> > >
> > > $$
> > > Y_q^{(\text{SA})} = \sum_{k} \operatorname{softmax}(A_{q,:}^{(1)})_k X_{k,:} W^{(1)} \in \mathbb{R}^{D_{out}}
> > > $$
> > >
> > > be the output of the single-head SA layer. Now, it is not hard to see that the equation $Y_q^{(\text{conv})} = Y_q^{(\text{SA})}$ does not have a general solution w.r.t the unknown parameters $W^{(1)} \in \mathbb{R}^{D_{in} \times D_{out}}$ and $\operatorname{softmax}(A_{q,:}^{(1)})_{:} \in \mathbb{R}^{H\times W}$. This follows by a degrees of freedom argument: on the convolutional layer, each $X_{k,:}$ within the kernel support (there are $K^2$ such pixels) is multiplied by a *different* $D_{in} \times D_{out}$ matrix $W_{q-k}$---thus, in total, we have $K^2 D_{in} D_{out}$ variables. On the SA side, each $X_{k,:}$ within the kernel support is multiplied by the *same* $D_{in} \times D_{out}$ matrix $W^{(1)}$ and the only thing that changes across $k$ are the non-zero scalar attention coefficients $\operatorname{softmax}(A_{q,:}^{(1)})_{k}$---yielding a total of $D_{in} D_{out} + K^2$ parameters (which is less than $K^2 D_{in} D_{out}$). It directly follows that a single-head SA layer cannot by-itself simulate all possible convolution layers.
> > >
> > >
> > >
> > >
> > > The figures presenting the computed attention probabilities show that some layers of MHSA (mainly 2 and 3) exploit relative positional encoding to attend to distinct pixels at a fixed shift from the query pixel reproducing the receptive field of a localized convolutional kernel. Other layers use the input content to compute attention probabilities (visible for the first layer of non-averaged attention probabilities, Figures 7-9 in Appendix). Interestingly, this aligns with the finding from (Bello et al. 2019) that convolution and self-attention combined outperform each taken separately. Such combination is learned in practice when optimizing an unconstrained fully-attentional model as shown in Figure 5.
> > >
> > > In light of the updated findings (with $W_{key},W_{qry} \neq 0$), we agree to nuance our claim that "learned fully-attentional models do behave similar to CNNs in practice" to "fully-attentional models learn to combine local behavior (similar to convolution) and global attention based on input content".
> > >
> > >
> > >
> > >
> > > As suggested by the reviewer, we have moved the experiment on learned positional encodings with $W_{key}=W_{qry}=0$ to Appendix to shorten the manuscript. The experiment section now covers:
> > >
> > > 1. the quadratic encoding (with $W_{key}=W_{qry}=0$) verifying that the theoretical construction can be learned,
> > > 2. a "pure" fully-attentional model similar to (Ramachandran et al., 2019) to study if some MHSA layers behave like CNN layers in practice.

---

### Official Review · AnonReviewer2 · 2019-10-24
**Official Blind Review #2**

**Rating:** 6

**Review:**

This paper studies the recent application of attention based Transformer networks for image classification tasks and asks the question as to the similarity of functions learned by these attention networks with the standard convolutional networks.

First the paper theoretically proves that a multi head self attention layer (appropriately defined for a 3 dimensional input) can represent a convolutional filter. The proof is based on constructing weights for the attention layers that results in a convolution operation. This construction uses rather crucially the relative positional encodings for the self attention layer. The paper claims that the results can be extended to other forms of positional encodings.

It looks like the construction is correct as far as I can tell. One caveat is that, It looks like, the weights of the attention layer need to be arbitrarily large (\alpha in Lemma 2) to exactly represent the convolution layer. I think this is not possible to avoid for exact representation. A comment on this after the results will be nice.

Finally the paper presents experiments on the Cifar10 dataset. The paper shows that the multihead attention units in the lower layers learn to attend on grid like structures on pixels, similar to a Conv filter.  I find the experiments to be nicely complementing the theoretical results, even though they are limited to the Cifar10 dataset.

Overall I think this paper takes a nice step towards understanding the similarities and differences between the Attention and Conv layers, and I suggest acceptance.

Minor:
First sentence in intro raise -> rise.


**Experience Assessment:**

I have published one or two papers in this area.

**Review Assessment: Checking Correctness Of Derivations And Theory:**

I assessed the sensibility of the derivations and theory.

**Review Assessment: Checking Correctness Of Experiments:**

I assessed the sensibility of the experiments.

**Review Assessment: Thoroughness In Paper Reading:**

I read the paper at least twice and used my best judgement in assessing the paper.

---

> ### Author Response · Authors · 2019-11-13
> **Answer to Official Blind Review #2**
>
> We thank the reviewer for their time assessing our work and their constructive feedback.
>
> Concerning the magnitude of $\alpha$, we distinguish two cases:
> 1. With infinite precision, the exact representation of one pixel (hard attention) requires $\alpha$ to be arbitrary large, despite that the attention probabilities of all other pixels converge exponentially to 0 as $\alpha$ grows.
> 2. With finite precision (i.e., in practice), the smallest positive float32 is approximately $10^{-45}$. As such, setting $\alpha=46$ is enough to obtain hard attention (which seems reasonable).
> Following your suggestion, we have added a remark in Section 3 to clarify this point.
>
> Thank you for pointing out the typo in the introduction.

---

### Author Response · Authors · 2019-11-15
**Authors' Comment about Experiment Datasets**

We would like to thank the reviewers again for their useful comments that allowed us to add some remarks on the theory and extend the experimental part. We believe that we addressed the raised concerns by updating the main text and conducting more experiments.

We would like to address one last point:
The reviewers pointed out through the discussion that our experiments merely cover CIFAR-10. We would like to argue that experiments on ImageNet/MSCOCO would not complement well the theory of the paper. Due to the quadratic memory footprint of the attentions scores (pixels to pixels), full Self-Attention cannot process large images without introducing spatial downsampling layers (ResNet backbone) and/or Local-Attention (Ramachandran et al., 2019). As the goal of the Section 4.3 is to study to what extent Self-Attention learns to behave like a CNN, enforcing locality to a 7x7 patch on 200x200 images as in (Ramachandran et al., 2019) already forces the model to adopt a receptive field similar to CNNs, hence defeating the purpose of our experiments.

We are also curious if our findings transfer to the Local-Attention setting but it is outside of the scope of our experimental section. However, we will include the visualization of Local-Attention scores in our interactive website once Ramachandran et al. (2019) release their model.

---

### Decision · Program_Chairs · 2019-12-19

**Decision:**

Accept (Poster)

**Comment:**

This paper studies the relationship between attention networks such as Transformers and convolutional networks. The paper shows that a special case of attention can be cast as convolution. However this link depends on using relative positional embeddings and generalization to other encodings are not given in the paper. The reviewers found the results correct, but we caution that the writing should better reflect the caveats of the approach.